# Brain Metastasis in Endometrial Cancer: A Systematic Review

**DOI:** 10.3390/cancers17030402

**Published:** 2025-01-25

**Authors:** Daniela Sambataro, Vittorio Gebbia, Annalisa Bonasera, Andrea Maria Onofrio Quattrocchi, Giuseppe Caputo, Ernesto Vinci, Paolo Di Mattia, Salvatore Lavalle, Basilio Pecorino, Giuseppa Scandurra, Giuseppe Scibilia, Danilo Centonze, Maria Rosaria Valerio

**Affiliations:** 1Medical Oncology Unit, Umberto I Hospital, 94100 Enna, Italy; annalisa.bonasera@asp.enna.it (A.B.); andrea.quattrocchi@asp.enna.it (A.M.O.Q.); giuseppe.caputo@asp.enna.it (G.C.); ernesto.vinci@asp.enna.it (E.V.); 2Department of Medicine and Surgery, Kore University, 94100 Enna, Italy; vittorio.gebbia@unikore.it (V.G.); paolo.dimattia@unikore.it (P.D.M.); salvatore.lavalle@unikore.it (S.L.); basilio.pecorino@unikore.it (B.P.); giuseppa.scandurra@unikore.it (G.S.); giuseppe.scibilia@unikore.it (G.S.); 3Surgery Unit, Umberto I Hospital, 94100 Enna, Italy; danilo.centonze@asp.enna.it; 4Diagnostic Imaging Department, Umberto I Hospital, 94100 Enna, Italy; 5Gynecology and Obstetrics Unit, Umberto I Hospital, 94100 Enna, Italy; 6Medical Oncology Unit, Cannizzaro Hospital, 95126 Catania, Italy; 7Gynecology Unit, Giovanni Paolo II Hospital, 97100 Ragusa, Italy; 8Medical Oncology Unit, Policlinic, University of Palermo, 90127 Palermo, Italy; mariarosaria.valerio@unipa.it

**Keywords:** endometrial carcinoma, brain metastasis, radiotherapy, stereotactic radiotherapy, target therapy, immune checkpoint inhibitor

## Abstract

The purpose of this study is to review the existing literature on the treatment of endometrial carcinoma with brain metastases. Brain metastases from endometrial cancer are rare and pose significant challenges in treatment, as no standardized approach exists. This study investigates the impact of different therapeutic strategies, including surgery, radiotherapy, systemic therapies, and their combinations, on patient survival. By evaluating these treatments in specific patient subgroups, such as those with solitary brain metastases or multiple brain metastases with extracranial disease, we aim to identify the most effective approaches to improve outcomes. Our findings may provide valuable insights to guide future research and inform clinical decision making for this challenging condition.

## 1. Introduction

Endometrial cancer is the sixth most common cancer among women and the second most frequent gynecological malignancy after cervical cancer, with over 400,000 new cases annually and 97,370 deaths reported in 2020 worldwide. Both the incidence and mortality of EC are rising, particularly in economically developed countries, likely due to the increasing prevalence of obesity and type 2 diabetes [1,2,3,4].

Most patients present with localized disease, with nearly 20% showing regional spread and 9% exhibiting distant metastases. According to the Surveillance, Epidemiology, and End Results (SEER) database and other studies, 5-year survival rates are stage-dependent. The 5-year survival rate in early-stage EC exceeds 95%, but it drops to 56–69% in patients with loco-regional spread and plummets to 17–20% in patients with distant metastases [5]. Recurrence occurs in approximately 20% of patients and typically presents in the pelvis and abdomen within 1–2 years of diagnosis [6,7,8]. Pelvic lymph nodes, such as internal and external iliac lymph nodes [9] and retroperitoneal lymph nodes, are the most common sites of metastasis. Distant metastases are less frequent, with the lung being the most common site (1.5%), followed by the liver (0.8%) and bones (0.6%), while the brain is the least frequent (0.2%). In patients with single-organ metastasis, the median overall survivals (OSs) for patients with lung, liver, bone, and brain metastasis were 11, 10, 8, and 5 months, respectively. Similarly, patients with lung, liver, bone, and brain metastasis had median cancer-specific survivals (CSS) of 14, 15, 11, and 8 months, respectively. [10]. Overall, 10–30% of all cancer patients develop BMs [11,12], which carry a poor prognosis, while in EC, the incidence is much lower, ranging from 0.2% to 1.4%.

In this paper, we review the fragmented medical literature on BMs in endometrial carcinoma.

## 2. Methods

This review was performed in accordance with the PRISMA (Preferred Reporting Items for Systematic Reviews and Meta-Analyses) guidelines and has not been registered. A comprehensive search was conducted in PubMed using the terms “brain metastasis and endometrial cancer” and their synonyms (cerebral, metastases, carcinoma, uterus, uterine), covering all available studies up to October 2024. Articles with extractable individual patient data were included. Case series lacking extractable individual data but containing relevant aggregated information were retained for quantitative analysis. Articles not written in English, French, Spanish, or Italian were excluded, as were those addressing unrelated topics based on predefined eligibility criteria. Non-pertinent records were excluded after review.

A total of 911 records were initially identified (850 from PubMed and 61 through citation searching). After screening for relevance, 748 records were excluded. Of the remaining 163 articles assessed for eligibility, 99 were included in the analysis. The PRISMA flow diagram (Figure 1) outlines the study selection process. Three studies utilizing data from the Surveillance, Epidemiology, and End Results database and the National Cancer Database (NCDB) were included solely for the purpose of assessing the incidence of brain metastases from endometrial carcinoma.

Since 1972, 594 cases of brain metastases from endometrial cancer have been reported in the literature.

All included studies were analyzed to extract the following variables: number of reported cases, patient age, stage at initial diagnosis, histological subtype, tumor differentiation grade, presence and location of extracranial metastases, time interval between the diagnosis of EC and the development of BMs, number and anatomical location of brain metastases, treatments administered, and survival from the time of BM diagnosis.

Three independent reviewers assessed each record and full-text article. The quality of single-case reports and case series was evaluated according to the Newcastle–Ottawa Scale for the domains applicable to the studies reported in the review as previously described [13]. Where reported, the following parameters were evaluated: the median age of patients, distribution of clinical stages at diagnosis, histological subtypes, differentiation grades, frequency and sites of extracranial metastases, and the number and distribution of BMs. Treatment modalities were analyzed, with median and range of survival calculated for each treatment type.

Patients were classified into four categories based on the number of brain metastases and the presence of extracranial metastases. For each category, treatment approaches were analyzed, and survival outcomes (medians and ranges) were reported.

Table 1 summarizes data extracted from the individual case reports of brain metastases arising from endometrial carcinoma. Additionally, one case of a patient with BMs from EC treated at our center was included, providing detailed demographic, clinical, and survival data.

This study was approved by the Ethics Committee of Kore University of Enna (the research project “Brain Metastasis of Endometrial Carcinoma” was registered under Prot. 26580; the ethical approval ensures adherence to ethical standards and guidelines for systematic reviews).

Table 1 summarizes the reported cases of brain metastasis from endometrial cancer.

## 3. Results

Eleven population studies showed a variable incidence of BM ranging from 0.2% to 0.97% [10,19,27,32,39,42,50,74,99,100,107]. One study focused solely on patients with stage I and II EC, reporting a 0.6% incidence of BMs [58]. One study evaluated only low-grade (G1–G2) patients and found an incidence of 0.75% [90]. Overall, most papers reported in this study were single-case reports (n = 62). Additionally, 37 papers reported small-numbered case series (n: 2–37), except for 3 papers reporting 132, 498, and 73 cases. The quality of the reports was poor (0–1 stars in the selection domain) in 22 cases (22%), fair (2 stars) in 28 cases (28%), and good in 49 cases (50%).

The median age of onset for BMs was 60 years (range: 33–82 years).

Among patients without metastases at diagnosis, 50% were at stage I, 15% at stage II, and 35% at stage III. Table 2 shows the histological types reported in the medical literature. The grading distribution was 11% G1, 23% G2, 66% G3.

In 43% of cases, the brain was the only site of metastasis; in the remaining 57%, metastases also involved the lungs, liver, lymph nodes, skin, peritoneum, bone, adrenal glands, and larynx.

In 66% of cases, BMs were supratentorial, 21% were infratentorial, and 13% involved both locations. The study population included four cases of pituitary metastases [22,25,89,96], three cases of leptomeningeal metastases [56,84], and, in one case, the metastases involved the dura mater [17].

In 49% of cases, BMs were solitary; in 5%, there were 2–3 metastases; and 46% had multiple MBs.

In 21% of patients, BMs were present at or preceded the diagnosis of EC. In the remaining patients, metastases occurred after a median of 18 months (range: 2–216 months) following the initial diagnosis of EC.

The median survival from the onset of BMs was 7 months (range: 0–171). In the group of patients with isolated BM, the median survival was 12 months (range: 1–171). Patients with 2–3 BM had a median survival of two months (range: 0.16–64). Patients with multiple BMs had a median survival of four months (range: 0.25–32). Patients with BMs in the absence of extracranial metastasis or with extracranial metastasis had median survivals of twelve months (range: 0.1–171 months) and 4.5 months (range: 0.25–84 months), respectively.

In patients with BMs as the first (primary) vs. subsequent site of recurrence (secondary), the median survivals were 9 months (range 0.1–171 months) and 5 months (range 0.25–108 months), respectively.

Treatment modalities are depicted in Table 3. Many reports were anecdotal, describing only 1 or 2 patients. Various treatments were used, including palliation, surgery, radiotherapy (RT), chemotherapy, stereotactic radiosurgery (SRS), or a combination of the above.

### 3.1. Isolated Brain Metastasis Without Extracranial Disease

Overall, 40% had isolated BM with no extracranial disease. In this group of patients, median survival was 13 months (range: 1–171). Surgery alone provided a median survival of eight months (range: 1–18); surgery and RT resulted in a median survival of 23 months (range: 2–118); radiotherapy or stereotactic radiotherapy combined with systemic therapy resulted in a survival of 30 months; surgery plus radiotherapy and systemic therapy led to a survival of 16 months (range: 6–74); and SRS with or without surgery resulted in a median survival of 15 months (range: 5–40).

One patient had stereotactic radiotherapy, and systemic therapy achieved a survival of 171 months [38]. Another patient with trimodal therapy, surgery, stereotactic radiotherapy, and systemic therapy achieved a survival of 38 months [41].

### 3.2. Isolated Brain Metastasis and Extracranial Disease

Twenty-three percent of patients had solitary BM and extracranial disease, with a median survival of 7 months (range: 1–84). RT alone resulted in a median survival of seven months (range: 1–38); surgical and RT yielded a median survival of 8.5 months (range: 4–84).

A trimodal treatment (surgery, RT, and systemic therapy) resulted in 34 months of survival in one patient [44]. A similar treatment approach in a patient with small cell carcinoma achieved 144 months of survival [78].

### 3.3. Multiple Brain Metastasis Without Extracranial Disease

Overall, 9% of patients had multiple BMs without extracranial disease, with a median survival of 6 months (range: 0.16–64). Most were treated with RT, yielding a median OS of five months (range: 0.16–17).

A patient with a serous carcinoma harboring a BRCA1 mutation was treated with niraparib after the progression of the disease with whole-brain radiotherapy (WBRT) plus temozolomide, achieving survival of 13 months from the onset of BMs [102].

### 3.4. Multiple Brain Metastasis with Extracranial Disease

Twenty-eight percent of patients had BMs and extracranial metastases, with a median survival of three months (range: 0–32). Most received RT, with a median OS of two months (range: 0.25–28); RT, systemic therapy, and surgery led to a median survival of 9 months; and radiotherapy or SRS and systemic therapy led a survival of 24 months.

A patient with EC with lung, liver, and peritoneal involvement, showing a BRCA1/2 wild type with a PTEN mutation, developed a BM 120 months after the initial diagnosis. After multiple treatments and SRS, she received olaparib, with a survival of eighteen months from the start of the treatment [59].

Another patient was treated with whole-brain radiotherapy and systemic therapy with pembrolizumab (PEM) and Lenvatinib (LEN) and achieved a survival of 32 months after a diagnosis of brain metastasis [111].

At our center, we treated a patient with endometrial carcinoma who underwent hystero-salpingo-oophorectomy followed by adjuvant chemotherapy. Upon mediastinal lymph node recurrence after 52 months, a new line of chemotherapy was initiated. Following further progression involving brain and lymph node metastases, the patient received stereotactic radiotherapy and subsequent systemic therapy with pembrolizumab and lenvatinib. Since the onset of brain metastases, the patient has achieved 30 months of survival and continues systemic treatment (Figure 2 and Figure 3).

## 4. Discussion

BMs from EC are rare, with our study showing an incidence ranging from 0.2% to 0.97%. Current scientific evidence supports the “seed and soil” theory proposed over a century ago [112]. The development of BMs is the result of multiple factors. The progressive accumulation of genetic alterations due to clonal evolution allows cells to metastasize [113]. Transcriptomic and epigenetic changes in newly established colonies allow micro-metastases to grow via metabolic adaptation [114]. Activating angiogenesis supports tumor growth and the production of chemokines and cytokines, which promote an immunosuppressive phenotype in resident immune cells [115,116,117].

Approximately 30% of endometrial carcinomas cases are associated with a germline mutation in mismatch repair (MMR) genes or show high microsatellite instability (MSI-H) [118].

The Cancer Genome Atlas was the first project to assess many tumors using whole-genome sequencing [119]. For EC, it identified four molecular subgroups: POLE-mutated tumors, MSI tumors, copy number low (CNL) tumors, and copy number high (CNH) tumors. The prognostic value of these molecular subgroups was evaluated in the PORTEC 3 study using univariate and multivariate analyses alongside clinicopathological factors, such as age, grade, lympho-vascular space invasion, and treatment. EC patients with abnormal expression of p53 had a poor prognosis, contrasting with the excellent survival outcomes of those with POLE-mutated EC, even in patients with high-grade and advanced-stage tumors. Patients with MMR-deficient or no specific molecular profile EC showed intermediate clinical outcomes [120]. Ashley et al. demonstrated that POLE and MSI subtypes of EC exhibit specific mutational processes, such as loss of proofreading by polymerase and MSI. At the same time, CNL and CNH tumors and uterine sarcomas are dominated by aging-related mutational processes [121]. Interestingly, while the molecular subtype of EC is generally stable between primary tumors and metastases, mutational signature changes occur in over 25% of cases, suggesting that further defects in DNA repair mechanisms may influence tumor progression [122].

Other factors, such as estrogens, prolactin, and pro-inflammatory adipokines, may intervene in EC progression. Estrogen, through its receptors and non-genomic interactions, induces remodeling of actin, a critical cytoskeletal protein, and the cell membrane. At a molecular level, this phenomenon depends on the induction of phosphorylation on Thr (558) of myosin, an actin-binding protein. This interaction enhances endometrial cells’ migration and implantation capacity, as demonstrated in preclinical studies on Ishikawa cell cultures and native endometrial stromal cells [122,123]. Other in vitro and in vivo studies have shown that estradiol promotes proliferation, migration, and invasion by activating the IL-6 pathway, which is involved in various signaling pathways associated with estrogen receptors (ERs), Bcl-2, Cyclin D1, and MMP2 [124].

The role of prolactin in EC metastasis growth has recently emerged, along with its association with reduced chemotherapy sensitivity. The mechanism of prolactin’s action is complex, involving endocrine, paracrine, and autocrine mechanisms, including interactions with immune cells. Prolactin exerts both receptor-dependent and receptor-independent effects. Its anti-apoptotic activity, mediated by prolactin receptors (PRLRs), involves blocking Stat5a/b expression, which increases anti-apoptotic Bcl-2 expression, decreases pro-apoptotic Bax expression, and upregulates Hsp90A. This latter chaperone protein protects cells from apoptosis. The proliferative effect of prolactin is further supported by its synthesis in vascular endothelial and stromal cells. Additionally, prolactin may interact with various agonist ligands through its receptor, creating a specific microenvironment in metastatic foci that promotes proliferative processes. Prolactin also directly stimulates vascular endothelial cell proliferation and indirectly increases the expression of VEGF and other proangiogenic factors [125,126]. Pro-inflammatory adipokines such as leptin, visfatin, and resistin are also implicated in the progression and spread of endometrial cancer cells [122]. Considerable attention has also been given to microRNA (miRNA) expression as a marker of metastatic risk in EC. miRNAs are small non-coding RNA fragments that physiologically regulate gene expression and are involved in oncogenesis and metastasis processes [127].

### Treatment

The prognosis of patients with BMs remains poor, with a median survival of 4 months following whole-brain radiotherapy and a 12% one-year survival rate [128,129]. However, survival beyond historical expectations has emerged due to innovative systemic therapies, such as targeted and immunotherapy. The most important prognostic variables are performance status, age, and systemic disease control. Prognostic variables also include the characteristics of BMs in terms of volume and number. Our study reports a seven-month median survival for patients with BMs from EC. Currently, no data have been reported regarding the possibility of early diagnosis of brain metastases. New studies correlating molecular evaluations with the onset of brain metastases could be helpful.

At this time, no standardized treatment pathway exists for these patients, as the available data are predominantly retrospective and based on small patient cohorts.

Local treatment options for BMs include surgical resection, WBRT, or SRS. Surgical resection is typically considered for patients with oligometastatic disease or significant mass effect and provides a histological diagnosis. Some randomized studies have demonstrated improved survival in patients with solitary BMs who undergo surgical resection followed by radiotherapy compared to surgery alone [130,131,132]. However, the potential benefits of surgery must be weighed against the risks and other prognostic factors.

In our review of published cases on the surgical treatment of BMs from EC, the median survival across the entire cohort was four months, and the median survival was eight months for patients with a single BM. For those who received surgery and radiotherapy, median survival increased to 15.5 months and 23 months for patients with a solitary BM and no extracranial involvement.

The reasons for the dismal results of surgery alone in patients with solitary brain metastases from endometrial cancer are not clear. It may be hypothesized that biological factors and potential sensitivity to radiation may influence results. Moreover, newer therapies such as immunotherapy and targeted therapies in combination could lead to better responses and survival, but further studies are needed.

WBRT remains a primary treatment option for BMs, with reported local control rates of approximately 80%. However, this benefit is often impaired by cognitive decline and deterioration of performance status [132,133,134].

WBRT is generally indicated for cases with multiple brain lesions or when palliative care is the only goal. Our review reports a 4-month median survival for patients treated with WBRT alone.

SRS involves the conformational delivery of high radiation doses to the target lesions, with a rapid dose fall-off at the lesion boundary [135,136]. Except for radioresistant tumors such as melanomas and sarcomas, SRS controls BMs without causing the cognitive decline associated with WBRT [137,138,139,140,141].

Brain metastases can evade the immune system through various mechanisms, including the secretion of immunosuppressive cytokines, downregulation of tumor-associated antigens (TAA) and major histocompatibility complex (MHC) class I expression, recruitment of regulatory T cells (Tregs) into the tumor microenvironment (TME) [142], or suboptimal functioning of host dendritic cells (DCs) [143]. SRS induces significant DNA damage in tumor cells, leading to cell death, while also activating multiple signaling pathways within the TME, inducing a pro-inflammatory state and potentially causing damage to surrounding stromal and endothelial cells [144]. Radiation has been shown to enhance the presentation of TAA by DCs to CD4+ and CD8+ T cells, thereby strengthening the immune system’s ability to recognize and target tumor cells [145]. It facilitates the maturation of antigen-presenting cells (APCs); enhances antigen–MHC complex assembly; and induces the secretion of critical inflammatory cytokines, including tumor necrosis factor-alpha (TNF-α), interferon-gamma (IFN-β), and chemokine ligand 16 (CXCL16). These cytokines attract immune cells to cross the blood–brain barrier (BBB) and infiltrate the TME [146].

Radiation can also regulate programmed death-ligand 1 (PD-L1) and cytotoxic T-lymphocyte-associated protein 4 (CTLA-4), potentially acting synergistically with immune checkpoint inhibitors.

Immunotherapy, particularly immune checkpoint inhibitors targeting PD-1/PD-L1 or CTLA-4 pathways, can potentiate this response by overcoming tumor-induced immune suppression. While specific data on endometrial cancer remain limited, studies in other tumor types, such as lung, melanoma, and cervical cancers, suggest that the combination can improve outcomes, including tumor control and survival. These findings provide a compelling rationale for exploring this approach in endometrial cancer, where the immune microenvironment and potential for neoantigen generation might similarly support the efficacy of combined modality therapy (Figure 4).

Table 4 depicts studies showing the treatment efficacy in treating BM from EC with SRS, but the retrospective nature and limited sample sizes do not allow for definitive conclusions.

Subsequently, the GOG 209 study confirmed the non-inferiority of the carboplatin-paclitaxel regimen compared to TAP, providing an alternative with a more favorable toxicity profile [148].

The integration of immunotherapy has further revolutionized treatment paradigms. The addition of pembrolizumab to carboplatin and paclitaxel reduced the risk of disease progression or death by 70% in the mismatch repair-deficient (dMMR) cohort and by 46% in the mismatch repair-proficient (pMMR) cohort, compared to placebo [149]. Similarly, the incorporation of dostarlimab into carboplatin-paclitaxel regimens resulted in a 2-year PFS rate of 36.1% and an OS rate of 71.3%, compared to 18.1% and 56.0%, respectively, in the placebo arm. In dMMR–MSI-H patients, 24-month PFS was 61.4% (95% CI, 46.3–73.4) in the dostarlimab group versus 15.7% (95% CI, 7.2–27.0) in the placebo group *p* < 0.001) [150].

Atezolizumab, combined with carboplatin and paclitaxel, demonstrated a 64% reduction in progression or death risk in dMMR patients (HR 0.36, 95% CI 0.23–0.57; *p* = 0.0005). In the overall population, median PFS improved to 10.1 months (95% CI, 9.5–12.3) with atezolizumab compared to 8.9 months (95% CI, 8.1–9.6) with placebo (HR 0.74, 95% CI, 0.61–0.91; *p* = 0.022) [151].

The DUO-E study explored the combination of durvalumab and paclitaxel-carboplatin with or without olaparib. In the intention-to-treat population, durvalumab significantly reduced the risk of progression or death compared to the standard regimen (HR 0.71, 95% CI, 0.57–0.89; *p* = 0.003). While olaparib added no notable PFS benefit in dMMR patients, its combination with durvalumab appeared advantageous in pMMR patients [152].

Therefore, PEM and dostarlimab have established efficacy as second-line treatment for patients with dMMR or MSI-high status [153,154].

Emerging evidence highlights the efficacy of targeted therapies in EC management. Lenvatinib, a tyrosine kinase inhibitor (TKI), selectively targets VEGFR, FGFR, RET, cKIT, and PDGFR, with preclinical studies confirming its antitumor potential, particularly in combination with immune checkpoint inhibitors [155,156,157,158].

Preclinical studies have demonstrated that inhibition of the FGFR pathway, alone or in combination with other signaling pathways or chemotherapy, induces antitumor activity in endometrial cancer models [159,160].

The phase-Ib/II KEYNOTE 146 trial demonstrated promising outcomes with LEN plus PEM, reporting a 38.0% objective response rate in metastatic EC patients without selection for microsatellite instability or PD-L1 status [161,162,163].

The phase-III KEYNOTE 775 trial compared LEN and PEM to physician’s-choice chemotherapy in patients previously treated for advanced EC. In pMMR patients, LEN and PEM significantly improved PFS, OS, and overall response rates. Median OS in pMMR patients was 18.0 months (95% CI, 14.9–20.5) compared to 12.2 months (95% CI, 11.0–14.1) with chemotherapy (HR 0.70, 95% CI, 0.58–0.83) [164,165].

Lastly, VEGF-A expression is heightened in brain metastases (BMs) compared to gynecologic primary tumors [77], while the active tumor immune microenvironment observed in both primary and metastatic sites supports combining antiangiogenic therapies with immune checkpoint inhibitors for the treatment of EC with BMs [103]. This therapeutic synergy warrants further exploration.

In our study, systemic therapy combined with radiotherapy resulted in a median survival of 10 months (range: 3–32 months). Trimodal treatment, comprising surgery, radiotherapy, and systemic therapy, achieved a median survival of 12 months (range: 4–74 months). Stereotactic radiotherapy combined with systemic treatment in three patients resulted in a median survival of 30 months (18,30,171 months). Two patients treated with radiotherapy or stereotactic radiotherapy in combination with pembrolizumab and lenvatinib showed median survivals of 32 and 30 months, respectively.

In patients with solitary brain metastases and no extracranial disease, radiotherapy or stereotactic radiotherapy combined with systemic therapy achieved a median survival of 30 months (range: 4–171 months). In the group of patients with multiple brain metastases and extracranial disease, radiotherapy or stereotactic radiotherapy combined with systemic therapy resulted in a median survival of 24 months (range: 3–32 months).

## 5. Conclusions

Our study has some limitations. Given the low incidence of BMs from EC, the literature reports retrospective studies and small case series, which support multimodal strategies such as radiotherapy, surgery, and systemic treatments. Surgery alone appears to have limited efficacy, even in patients with single metastases and no extracranial disease. The addition of radiotherapy improves survival in patients with solitary brain metastases and no extracranial disease. Systemic treatment with tyrosine kinase inhibitors, immunotherapy, and radiotherapy appears to be a promising therapeutic approach.

In conclusion, we report a detailed review of the available medical literature, even anecdotal cases. Therefore, the data reported above should be interpreted with caution. Patients’ clinical characteristics probably influenced physicians’ treatment choices rather than following reliable therapy guidelines.

## Figures and Tables

**Figure 1 cancers-17-00402-f001:**
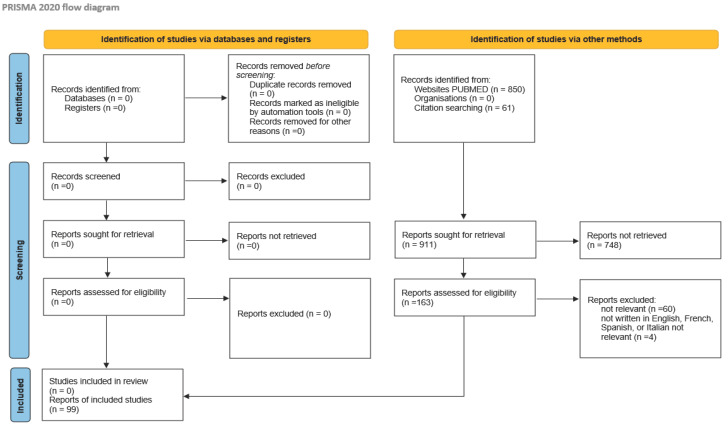
Literature review according to PRISMA 2020 method.

**Figure 2 cancers-17-00402-f002:**
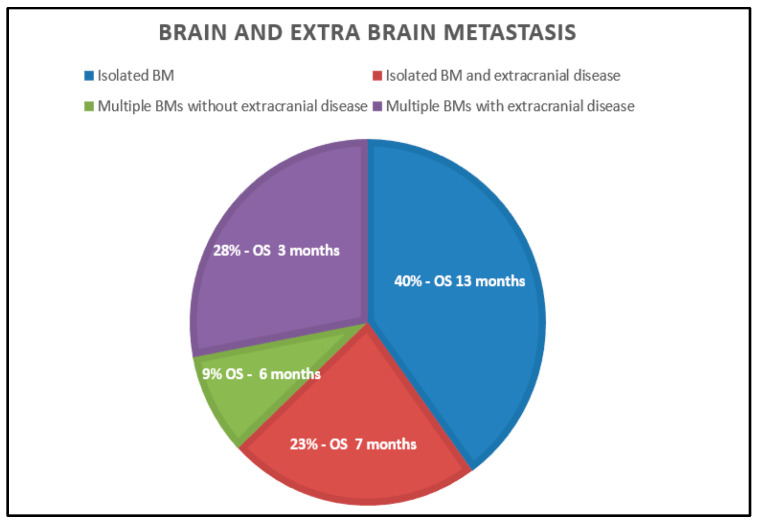
Median survival in patients with isolated brain metastasis and no extracranial disease, isolated brain metastasis and extracranial disease, multiple brain metastasis and no extracranial disease, and multiple brain metastasis and extracranial disease.

**Figure 3 cancers-17-00402-f003:**
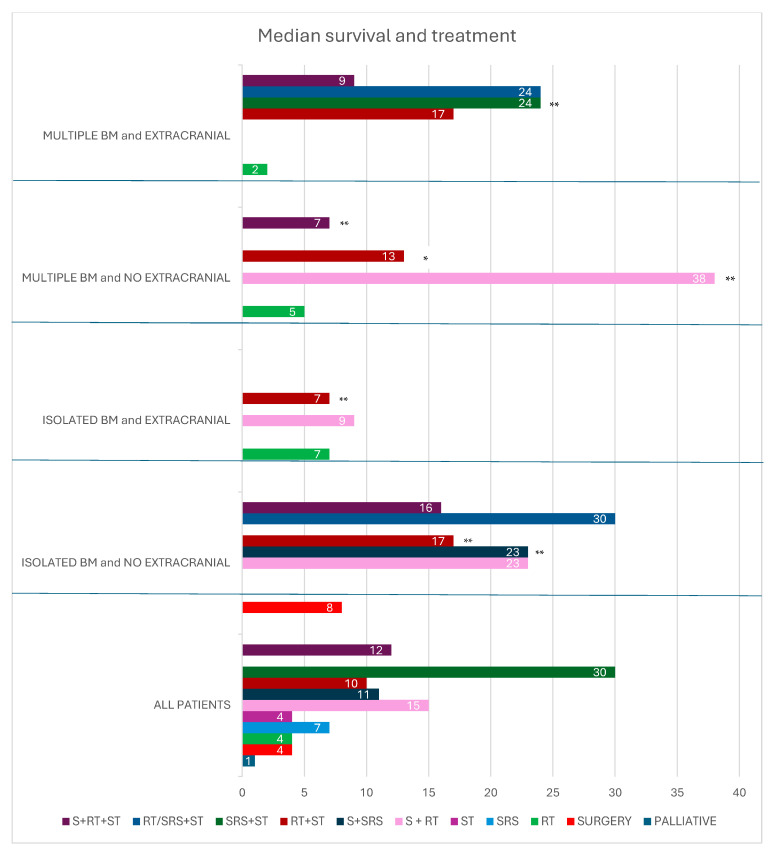
Median survival and treatment in all patients, patients with isolated brain metastasis and no extracranial disease, isolated brain metastasis and extracranial disease, multiple brain metastasis and no extracranial disease, and multiple brain metastasis and extracranial disease. * 1 patient; ** 2 patients. BM brain metastasis; RT radiotherapy; S surgery; SRS stereotactic radiotherapy; ST systemic therapy.

**Figure 4 cancers-17-00402-f004:**
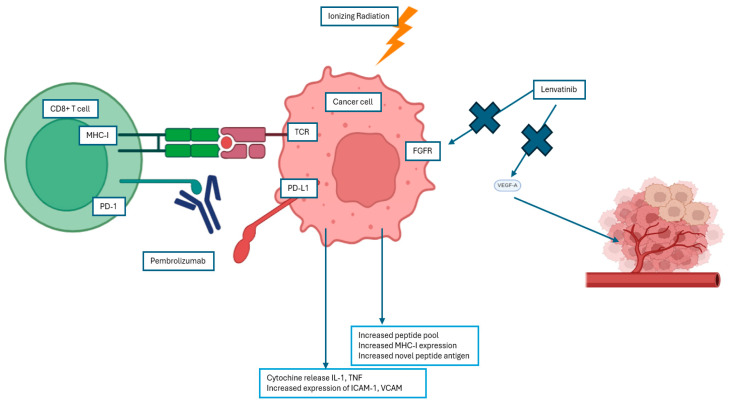
Effect of radiotherapy, immunotherapy, and TKI inhibitor. CD8+ T cell: cytotoxic T lymphocyte, MHC-I: major histocompatibility complex class I, TCR: T cell receptor, PD-1: programmed cell death protein 1, PD-L1: programmed death-ligand 1, FGFR: fibroblast growth factor receptor, IL-1: Interleukin 1, TNF: tumor necrosis factor, ICAM-1: intercellular adhesion molecule 1, VCAM: vascular cell adhesion molecule 1.

**Table 1 cancers-17-00402-t001:** Cases of brain metastases from endometrial carcinoma.

Author, Year [Ref]	No. of Patients EC BM/Total	Age at BM Year (Range)	Stage at Diagnosis (n. pt)	Histology (n. pt)	Grade (n. pt)	Other Metastasis (n. pt)	Interval EC to BM Months	Site of BM (n. pt)	No. of BM (Range)	Treatment for BM (n. pt)	Survival After BM Months (Range)
Salibi B.S., Beltaos E., 1972 [14]	1	63	NS	adenocarcinoma	NS	no	6	ST	single	S	>18
Nakano K.K. et al., 1975 [15]	1	77	NS	clear cell	NS	lung, adrenal, mediastinum	26	ST	single	S + RT	4
Hacker R.J., Fox J.L., 1980 [16]	1	80	/	adenocarcinoma	NS	no	216	IT	single	S	1
Turner D.M., Graf C.J, 1982 [17]	1	83	NS	adenocarcinoma	G3	ns	NS	dura mater	single	S	1
Kishi K. et al., 1982 [18]	1	NS	NS	NS	NS	NS	NS	ST	single	NS	>96
Aalders J.G. et al., 1984 [19]	11	NS	NS	NS	NA	no (8) yes (3)	NS	NS	NS	NS	NS
Ritchie W.W. et al., 1985 [20]	1	61	IIIC	C undifferentiated	G3	larynx	NS	ST	multiple	NS	4
Savage W. et al., 1987 [21]	1	70	I	adenocarcinoma	G1	yes	0	ST	single	RT	14
McCormick P.C. et al., 1989 [22]	1/4 ##	64	NS	NS	NS	lung	36	pituitary	single	S	NS
Sawada M. et al., 1990 [23]	1	43	IIIC	C undifferentiated	G3	obturator nodes	2	ST	single	S + RT	84
Brezinka C. et al., 1990 [24]	1	64	IC	adenocarcinoma	G2	abdomen, chest nodes	1	ST	single	S	1
Lieschke G.J. et al., 1990 [25]	1	NS	IVB	adenocarcinoma	NS	no	NS	pituitary	single	PALLIATIVE	1
Kottke-Marchant K. et al., 1991 [26]	1	59	IIIA	clear cell	G3	paraaortic nodes	before	ST	single	biopsy + RT	38
	1	43	IIIA	endometrioid	G3	no	0	ST/IT	2	S	0.75
	1	46	IA	endometrioid	G3	no	before	ST	single	S + RT	9
De Porre, Subandono Tjokrowardojo, 1992 [27]	11 data for 1	67	NS	adenosquamous	G3	no	8	ST	single	S	14
Thomas H, Lambert H.E., 1992 [28]	1/3 #	51	I	adenocarcinoma	NS	no	24	ST	single	S + RT	>84
Wroński M. et al., 1993 [29]	1	70	/	serous	/	lung	20	IT	single	RT	5.5
	1	60	/	clear cell	/	lung	84	/	2	RT	1.5
Iqbal J.B, Ironside J.W., 1993 [30]	1	58	NS	carcinosarcoma	/	no	8	ST	single	S + RT	>25
Ruelle A. et al., 1994 [31]	1	64	NS	adenocarcinoma	G3	lung, bone	14	IT	single	S + RT	9
	1	63	I	adenocarcinoma	G3	paraaortic nodes	before	ST	single	S + RT	>24
Cormio G. et al., 1996 [32]	1	59	IV	endometrioid	G3	pelvis lung	11	ST	multiple	PALLIATIVE	1
	1	57	IB	endometrioid	G3	no	13	ST	single	S + RT	83
	1	68	IB	adenosquamous	G2	no	35	IT	multiple	RT	3
	1	49	IIIA	endometrioid	G3	lung, bone	81	ST	single	PALLIATIVE	2
	1	57	IB	endometrioid	G1	no	36	ST	single	S + RT	28
	1	57	IC	endometrioid	G2	no	17	ST	single	S	3
	1	65	IIA	endometrioid	G3	no	8	ST	single	PALLIATIVE	1
	1	47	IV	clear cell	G3	lung	3	ST/IT	multiple	PALLIATIVE	1
	1	51	IIIC	endometrioid	G1	lung, liver	46	ST/IT	multiple	PALLIATIVE	1
	1	63	IC	endometrioid	G3	no	58	ST	single	PALLIATIVE	1
De Witte O. et al., 1996 [33]	1/4 #	67	IA	adenocarcinoma	NS	no	24	ST	single	S + RT	60
	1/4 #	40	IVB	adenosquamous	G3	no	before	ST	single	S + RT	NED 2
Salvati M. et al., 1998 [34]	1	48	IA	adenocarcinoma	G3	no	10	ST	single	S + RT + ST	20
	1	54	IA	adenocarcinoma	G3/4	no	26	ST	single	S + RT + ST	74
Martinez-Manas R.M. et al., 1998 [35]	1	76	IIB	adenocarcinoma	G3	no	18	ST	single	S	8
Ogawa K. et al., 1999 [36]	1	43	IIB	adenocarcinoma	G2	lung, paraaortic nodes	36	ST	multiple	RT	5
	1	64	IIB	adenocarcinoma	G3	Chest and supraclavicular nodes, adrenal gland	18	ST/IT	multiple	RT	3
Crispino M. et al., 2000 [37]	1	57	IC	NS	G3	no	12	ST	single	S + RT	3
Petru E. et al., 2001 [38]	1	59	IVA	adenocarcinoma	G3	no	before	ST	single	SRS + ST	171
	1	69	IIIC	serous	G3	no	before	IT	single	SRS	15
Mahmoud-Ahmed A.S. et al., 2001 [39]	1	45	IVB	adenocarcinoma	NS	no	0	ST/IT	multiple	RT	6
	1	61	IVB	adenocarcinoma	NS	bone	0	ST/IT	multiple	RT	1
	1	54	IIIC	adenocarcinoma	NS	no	12	ST	single	S + RT	11.5
	1	67	IB	adenosquamous	NS	bone	21	ST	multiple	RT	2
	1	44	IIIB	adenosquamous	NS	bone, liver, lung	4	ST/IT	multiple	RT	0.25
	1	66	IIIA	adenocarcinoma	NS	lung	2	ST	single	S + RT	15.5
	1	46	IIIC	adenocarcinoma	NS	peritoneum	70	ST	multiple	RT	3
	1	44	IVB	adenocarcinoma	NS	no	1	ST	multiple	S	1.5
	1	48	IIIB	adenosquamous	NS	nodes	14	ST	multiple	S	4
	1	65	IVB	adenocarcinoma	NS	bone	3	IT	single	S + RT	15
Sewak S. et al., 2002 [40]	1	63	IB	endometrioid	G3	chest	48	IT	single	S + RT	6
Shiohara S. et al., 2003 [41]	1	48	IIIA	adenosquamous	G3	no	0	ST	single	S + SRS + ST	38
Gien L.T. et al., 2004 [42]	8	67 (48–82)	IIB (2) IIIC (4) IVB (2)	adenocarcinoma (4) serous (2) clear cell (1) adenosquamous (1)	G2 (3) G3 (5)	no (2) yes (6)	0 (2) metachronous (6)	ST (4) IT (2) ST/IT (2)	single (4) multiple (4)	RT (6) ST (1) PALLIATIVE (1)	3.5 (0.25–7)
Elliott K.S. et al., 2004 [43]	1	51	IIB	adenocarcinoma	G3	no	2	ST	single	S + RT + ST	>30
Salvati M. et al., 2004 [44]	1	62	IA	adenocarcinoma	NS	no	48	ST	single	S + RT	>9
	1	51	IVB	adenocarcinoma	G3	yes	0	ST	single	S + RT + ST	34
N’Kanza A.L. et al., 2005 [45]	1	61		carcinosarcoma		chest, abdomen, pelvis	0	ST/IT	2	RT	2
Ota T. et al., 2005 [46]	1	34	IA	adenocarcinoma	G1	ovarian, liver	60	NA	ns	NS	11
Dietrich C.S. et al., 2005 [47]	1	72	IC	NS		liver, lung	22	NS	ns	NS	2
Lee W.J. et al., 2006 [48]	1	54	IB	endometrioid	G3	no	108	ST	2	RT	0.16
Llaneza-Coto A.P. et al., 2006 [49]	1	43	IIA	adenocarcinoma	G3	no	0	ST	single	S	1
Chura J.C. et al., 2007 [50]	20	64 (49–78)	IA (1) IB (2) IIIA (4) IIIC (4) IVB (9)	adenocarcinoma (11) carcinosarcoma (3) adenosquamous (2) serous (1) C undifferentiated (3)	G1 (3) G2 (6) G3 (11)	yes (16) no (4)	11.5 (0.6–73.6)	NS	Single (8) multiple (12)	RT (7) RT + ST (4) S + RT (1) SRS + RT (1) S + RT + ST (3) PALLIATIVE (4)	2 (0.1–39)
Orrù S. et al., 2007 [51]	1	61	IIIC	adenocarcinoma	G3	no	17	ST	2	S + RT	>64
	1	60	IIIA	adenocarcinoma	G3	no	6	ST	single	RT	4
	1	49	IIIB	adenocarcinoma	G3	no	10	ST	single	S + RT	>16
Sohaib S.A. et al., 2007 [52]	1	NS	NS	adenocarcinoma	NS	no	NS	NS	single	NS	NS
Monaco E. et al., 2008 [53]	6/27 #	60.4	NS	NS	NS	NS	NS	NS	NS	SRS ± S ± RT ± ST	7 (0.2–25)
Ramirez C. et al., 2008 [54]	1	61	IIB	adenocarcinoma	G3	no	12	IT	multiple	RT	17
Al Mujani A. et al., 2008 [55]	1	69	NS	adenocarcinoma	NS	lung	83	ST	multiple	NS	NS
Asensio S. et al., 2009 [56]	1/3 #	72	/	serous	G3	no	/	leptomeninges	single	RT + ST	4
Srikantia et al., 2009 [57]	1	41	IC	endometrioid	G3	Liver, bone, subcutaneous, spine	3	ST	multiple	RT + ST	NS
Blecharz P, et al., 2011 [58]	10	/	/	/	/	/	/	/	/	S + RT (2) RT (8)	2/10 alive at 36 m
Forster M.D et al., 2011 [59]	1	59	NS	adenocarcinoma	NA	lung, liver, peritoneum	120	ST	multiple	SRS + ST *	18
Yamashita S. et al., 2011 [60]	1	57	NS	NS	NA	no	0	ST	single	ST	>3
Menendez J.Y. et al., 2012 [61]	5/36 ##	/	/	/	/	/	/	/	/	SRS	/
Cabuk-Comert E et al., 2012 [62]	1/12 #	53	IA	adenocarcinoma	G2	peritoneum	29	IT	multiple	RT + ST	>30
	1/12 #	54	IIIA	serous	G3	peritoneum	2	IT	single	ST	5
Talwar S. and Cohen S, 2012 [63]	1	63	IA	serous	G3	lung	63	NS	multiple	RT + ST	>3
Berretta R. et al., 2013 [64]	1	67	IVB	C undifferentiated	G3	adrenal gland, pelvis	0	IT	multiple	S + ST	4
Gulsen S. and Terzi A., 2013 [65]	1	71	NS	serous	G3	no	27	ST/IT	multiple	S + RT + ST	>9
Bergamini A. et al., 2014 [66]	1	46	IA	endometrioid	G2	no	2	IT	single	S + SRS	40
	1	56	IA	endometrioid	G3	no	2	IT	single	S + SRS	5
Nassir M. et al., 2014 [67]	1	72	II	adenocarcinoma	G2	no	27	ST	single	S + SRS + RT	>14
Shepard M.J. et al., 2014 [68]	6/16 #	/		NS	/	/	25	/	single	S + RT (1) SRS (5)	8.3 m
Gressel G.M. et al., 2015 [69]	21/47 #	56	I (1) II (2) III (6) IV (12)	serous 4 endometrioid (13) squamous (1) adenosquamous (1)	/	yes (17) no (4)	9.5	NS	Single (5) multiple (16)	S (1) RT (15) S + RT (2) PALLIATIVE (4)	30 4 (0–123) 26 (7–45) 4 (2–10)
Kim Y.Z. et al., 2015 [70]	19/61 #	58	II (3) III (8) IV (8)	endometrioid (6) adenocarcinoma (11) leiomyosarcoma (2)	/	yes (8)	NS	ST (13) IT (6)	single (11) multiple (8)	S (9) RT (14) PALLIATIVE (2) ST (9)	23 (17.8–28.8)
Kouhen F. et al., 2015 [71]	1	62	IA	adenocarcinoma	G3	no	24	ST	single	RT + ST	30
Narasimhulu D.M. et al., 2015 [72]	1	81	IA	serous	G3	lung, vulvar	36	ST	2	RT	4
	1	62	IA	adenocarcinoma		vaginal, para-aortic nodes	24	ST	single	S	8
Sinai, 2013 [73]	1	55	IIIC2	serous	G3	no	11	ST	single	S + RT + ST	>12
Sierra T. et al., 2015 [73]	1	55	IVB	serous	NS	lung	11	ST	multiple	S + RT + ST	>8
Uccella S. et al., 2016 [74]	1	66	IIIC	serous	G3	bone, lung	18	ST	single	SRS	6
	1	77	IA	adenocarcinoma	G2	no	57	IT	single	S + RT	50
	1	55	IIIC	adenocarcinoma	G3	no	5	ST	single	S partial + RT	7
	1	54	IB	adenosquamous	G3	no	1	ST	single	S + RT	12
	1	65	IA	adenocarcinoma	G3	no	6	ST	single	S + RT	>64
	1	63	IVB	serous	G3	no	0	ST/IT	2	S partial + RT	5
	1	74	IB	adenocarcinoma	G1	retro-crural nodes	40	ST	single	S + RT	8
	1	62	IIIA	adenocarcinoma	G3	no	3	ST	single	S + RT	NED 118
	1	65	IVB	serous	G3	abdominal	19	ST	multiple	RT	17
	1	60	IVB	C undifferentiated	G3	abdominal	5	ST	3	PALLIATIVE	0
	1	79	IIIA	C undifferentiated	G3	liver, para-aortic nodes, lung	5	ST	single	PALLIATIVE	1
	1	42	IVB	adenocarcinoma	G3	no	0	IT	single	S + RT	NED 100
	1	78	IIIC	adenocarcinoma	G2	pelvic nodes	4	ST	2	RT	1
	1	74	IVB	adenocarcinoma	G3	lung	4	ST/IT	multiple	RT	5
	1	80	IIIA	adenocarcinoma	G3	lung, liver	13	ST/IT	multiple	RT	2
	1	62	IIIA	adenocarcinoma	G1	peritoneum	5	ST	2	PALLIATIVE	0.5
	1	52	IVB	adenocarcinoma	G3	peritoneum, bone, neck	7	ST	multiple	RT + ST	3
	1	59	IB	adenocarcinoma	G3	lung	1.5	ST	multiple	RT	28
Shin H.K. et al., 2016 [75]	6/24 #			Endometrioid (5) small cell (1)	/	/	/	/	/	SRS	7.5 (2–51)
Matsunaga S. et al., 2016 [76]	37/70 #	/	/	/	/	/	/	/	/	SRS	8
Divine L.M. et al., 2016 [77]	32/100 #	57 (28–82)	/								19
Sawada M. et al., 2016 [78]	1	40	IVB	small cell carcinoma	/	liver	0	IT	single	S + RT + ST	144
Keller A. et al., 2016 [79]	10/33 #	/	/	/	/	/	/	/	/	SRS ± RT	6
Gilani M.A. et al., 2016 [80]	6/23 #	63	I (2) III (2) unknown (2)	adenocarcinoma (3) sarcoma (3)	/	/	22	/	/	/	2
Könnecke HK et al., 2016 [81]	1	55		carcinosarcoma		no	3	ST	multiple	S + RT + ST	4
	1	67		carcinosarcoma		no	14	IT + medullary	multiple	S + RT	12
Takeshita S. et al., 2017 [82]	12/47 #	73 (56-80)	I (4) III (5) IV (3)	serous (3) endometrioid (6) carcinosarcoma (2) mixed-epithelial C (1)		yes (11)	22 (6–148)	/	multiple (5) single (3) unknown (4)	S + RT (3) RT (4) RT + ST (1) PALLIATIVE (4)	PFS 9 (2–36)
Takeshita S. et al., 2017 [82]	1					yes			single	RT	22
	1					no			single	S + RT	NED 52
Kimyon G. et al., 2017 [83]	1	69	/	/	/	no	/	NS	single	S + RT	21
Toyoshima M. et al., 2017 [84]	2/8 #	62	/	/	/	/	/	leptomeninges	single	PALLIATIVE	1.5
Kasper E. et al., 2017 [85]	1/8 #	/	/	/	/	/	/	/	/	SRS	/
Healy V. et al., 2017 [86]	1	74	IIIA	carcinosarcoma		no	16	IT	single	S + RT	>2
Johnston H. et al., 2017 [87]	6/33 #	/	/	/	/	/	/	/	/	SRS ± RT	6
Stamates M.M. et al., 2018 [88]	1	60	IIIC2	NS	/	no		IT	single	S + RT + ST **	>12
Salvatore B. et al., 2018 [89]	1	77	/	/	/	/	/	pituitary	/	/	/
Cybulska P. et al., 2018 [90]	23	66 (47–86)	I 15 II 2 III 3 IV 3	NS	G1 (9) G2 (14)		29.7	IT (6)	single (6) Multiple (17)	S + ST (2) S + RT + ST (4) S + RT (3) RT + ST (1) RT (7) PALLIATIVE (6)	5.1 (2.2–8.1)
Eulálio Filho WMN et al., 2019 [91]	1	56	IB	serous	G3	no	16	ST	single	S + RT	12
Zhang Y. et al., 2019 [92]	24/42 #	61.2	I 3 II 2 III 12 IV 7	endometrioid G 1/2 (4) endometrioid G 3 (7) serous (4) carcinosarcoma (3) adenosquamous (1) clear cell carcinoma (1) leiomyosarcoma (1) pleomorphic sarcoma (1)	G1 (2) G2 (2) G3 (20)		1.9	NS	Single (8) multiple (16)	NS	NS
Moroney M.R. et al., 2019 [93]	1	61	IB	adenocarcinoma	G3	lung pelvis	20	IT	single	S + RT	7
	1	66	II	serous	G3	lung	32	ST	single	RT + ST	3
	1	50	IA	adenocarcinoma	G2	pelvis, peritoneum	34	ST	2	PALLIATIVE	3
	1	55	IVB	adenocarcinoma	G3	lung, pelvis, peritoneum, bone	7	ST	single	RT	3
	1	71	IVB	adenocarcinoma	G3	lung, bone	20	ST	single	RT + ST	10
	1	49	IIIC	adenocarcinoma	G3	lung pelvis	57	ST	multiple	RT	2
	1	45	IVB	adenocarcinoma	G3	lung bone	9	ST	multiple	PALLIATIVE	1
	1	54	IVB	serous	G3	lung pelvis, peritoneum,	12	ST	multiple	S + RT + ST	>12
	1	82	IIIC	adenocarcinoma	G3	lung peritoneum	82	ST	single	RT	7
	1	51	IA	adenocarcinoma	G2	lung pelvis, peritoneum,	199	ST	single	RT	1
	1	51	IA	adenocarcinoma	G2	lung peritoneum	37	ST	multiple	S + RT + ST	9
	1	33	IIIB	adenocarcinoma	G1	lung	110	ST	single	S + RT	6
Yang F. et al., 2019 [94]	1	64	IA	adenocarcinoma	G2	no	156	ST	single	S	>12
Katiyar V. et al., 2019 [95]	1	41	IV	carcinosarcoma	G3	Lung, nodes	7	ST	Single	RT	1
Mao W. et al., 2020 [10]	132		NS	endometrioid (75) serous (3) carcinosarcoma (12) mixed epithelial (15) undifferentiated (11)	G1 (6) G2 (13) G3 (45) unknown (30)	yes (64) no (41)		NS	NS	NS	1–5 10% at 3 and at 5 y
Du H. et al., 2020 [96]	1	68	IVB	endometrioid	G1	no	0	pituitary	single	PALLIATIVE	3
Ogino A. et al., 2020 [97]	12/37 #	54	/	NS	/	/	22.5	/	3	S + RT (2) RT (10)	6 m
Guo J. et al., 2020 [98]	27	/	/	/	/	/	/	/	/	/	6
Nasioudis D et al., 2020 [99]	498/853 #	61	/	/	/	/	/	/	single (165) multiple (333)	ST (228) RT (169) SRS (44)	4.34 m
Bhambhvani H.P. et al., 2021 [100]	30	62 (39–79)	I (6) II (1) III (11) IV (9)	endometrioid (16) serous (7) carcinosarcoma (5) glassy cell C (1) clear cell 1	G1 (3.3%) G3 (80%)	yes 21 no 9 lung 70% bone 36.7% liver 20%	21 (1.4–134)	ST (16) IT (6) ST/IT (8)	2 (1–20)	S + SRS (11) SRS (17)	6.8 (1–58.2) 15.7 (2.8–58.2) 5.6 (1–50.4)
Beucler N. et al., 2021 [101]	1	70	IVA	endometrioid	G2	no	38	ST	single	S + RT	108
	1	53	II	adenocarcinoma		yes	30	ST	single	S + SRS	>6
Wang Q. et al., 2021 [102]	1	62	NS	serous BRCA 1 mut	G3	no	16	NS	multiple	RT + ST ***	13
Gill M.C. et al., 2021 [103]	21	/	/	/	/	/	18	/	/	/	6.8 m (2.6–14.7)
Leung S.O.A et al., 2021 [104]	1	64		endometrioid	G3	no	0	ST + IT	single	S + RT + ST	6
Crain N. et al., 2021 [105]	1	57	IIIA	serous	G3	no	72	ST	Single	S + RT	>14
Karpathiou G. et al., 2022 [106]	6/18 #	63		endometrioid	G2 (1) G3 (5)	no	27.8	ST (5) IT (1)	single single	NS NS	5
Zhang M. et al., 2023 [107]	121	/	/	non endometrioid 83%	/	/	22.8	NS	(2–10)	NS	NS
Wei Z. et al., 2023 [108]	25/50 #	65.5	/	/	/	/	/	/	/	SRS	/
Matsunaga S. et al., 2023 [109]	73/134	/	/	/	/	/	/	/	/	SRS	/
Butorac D. et al., 2024 [110]	1						0		single		
Shelvin K.B. et al., 2024 [111]	1	53	IIIC2	serous	G3	Lung, liver	33	ST/IT	multiple	RT + ST ****	32
Sambataro D, personal communication, 2024	1	74	II	endometrioid	G1	mediastinal nodes, interaortocaval nodes	52	ST/IT	2	SRS + ST ****	30

before = the diagnosis of BM precedes the diagnosis of EC; BM, brain metastasis; C, carcinoma; EC, endometrial cancer; NED, no evidence of disease; NS, non-specified; IT, infratentorial; n. pt, number of patients; RT, radiotherapy; SRS, stereotactic radiotherapy; S, surgery; ST (site), supratentorial; ST (treatment), systemic therapy; * Olaparib, ** Tamoxifen, *** Niraparib; **** Pembrolizumab/Lenvatinib; # gynecological cancer; ## various cancers.

**Table 2 cancers-17-00402-t002:** Histology of endometrial carcinomas.

Histology	Rate
Adenocarcinoma NOS *	34.5%
Endometrioid adenocarcinoma	28.5%
Serous carcinoma	15%
Carcinosarcoma	7%
Adenosquamous carcinoma	4.8%
Clear cell carcinoma	2.5%
Undifferentiated carcinoma	3%
Mixed endometrial carcinoma	1 case
Glass cell carcinoma	1 case
Squamous carcinoma	1 case
Small cell carcinoma **	2 cases
Leiomyosarcoma **	3 cases
Pleomorphic sarcoma **	1 case
Sarcoma **	1 case

* NOS: not otherwise specified; ** not evaluated in the analysis.

**Table 3 cancers-17-00402-t003:** Treatment modalities.

Reference	Treatments	Percent of Patients	Median Survival	Range
			(Months)	
[25,32,42,50,69,70,74,82,84,90,93,96]	Palliative therapy	8%	1	0.5–4
[14,16,17,22,24,26,27,32,35,39,49,69,70,72,94]	Surgery	5%	4	0.75–30
[21,26,29,32,36,39,42,45,48,50,51,54,57,69,70,72,74,82,90,93,95,97]	Radiotherapy	23%	4	0.16–38
[38,61,68,74,75,76,85,100,108,109]	SRS	38%	7	1–50
[42,60,62,70]	Systemic therapy	11% *	3,5	Na
[15,23,26,28,30,31,32,33,37,39,40,44,50,51,58,68,69,74,81,82,83,86,90,91,93,97,101,105]	Surgery + radiotherapy	10%	15	2–118
[66,100,101]	Surgery + SRS	10%	11	5–58
[50,56,57,62,63,71,74,82,90,93,102,111]	Radiotherapy and systemic therapy	4%	10	3–32
Sambataro D. personal communication [38,59]	SRS and systemic therapy	3 patients	30	18–171
[34,43,44,50,65,73,78,81,88,90,93,104]	Surgery, radiotherapy and systemic therapy	5%	12	4–74
[64,90]	Surgery and systemic therapy	3 patients **	4	Na
[67]	Surgery, SRS and radiotherapy	1 patient	14	Na
[41]	Surgery, SRS and systemic therapy	1 patient	38	Na

* 2 patients evaluable for survival; ** 1 patient evaluable for survival. SRS: stereotactic radiotherapy surgery; Na: not applicable.

**Table 4 cancers-17-00402-t004:** Efficacy of stereotactic radiosurgery in endometrial cancer.

Author, Year [Ref]	N. PATIENTS/DESEASE	TREATMENT	EFFICACY
Petru E. et al., 2001 [38]	2/endometrial cancer	SRS + S + ST SRS	Survival 171 months Survival 15 months
Shiohara S. et al., 2003 [41]	1/endometrial cancer	S + SRT + ST	Survival 38 months
Chura J.C. et al., 2007 [50]	1/20 endometrial cancer	SRS + WBRT	Median survival of 20 patients 2 months (range 0.1–39 months)
Monaco E. et al., 2008 [53]	6/endometrial cancer 21/ovarian cancer	SRS 100% S 14.8% WBRT 66.7% ST 88.9%	Median survival 7 months
Forster M.D. et al., 2011 [59]	1/endometrial cancer	SRS + CT	Median survival 18 months
Menendez J.Y. et al., 2012 [61]	4/sarcoma 2/prostate cancer 3/thyroid cancer 5/endometrial cancer 7/ovarian cancer 2/cervical cancer 6 /esophageal cancer 2/bladder cancer 1/liver cancer 1/pancreatic cancer 3 /testicular cancer	44 gamma knife sessions treating 74 tumors	63 tumors showed no radiographic evidence of progression, 13 tumors demonstrated radiographic progression between one and 12 months after gamma knife treatment
Bergamini A. et al., 2014 [66]	2/endometrial cancer	SRS + WBRT	survival 40 and 5 months
Nassir M. et al., 2014 [67]	1/endometrial cancer	S + SRT + WBRT	alive 14 months
Shepard MJ et al., 2014 [68]	8/ovarian cancer 6/endometrial cancer 1/cervical cancer 1/leiomyosarcoma	median dose to the tumor margin was 20 Gy (range 10–22 Gy), and the median maximum radio-surgical dose was 31 Gy (range 16–52.9 Gy)	ovarian cancer median survival 22.3 months endometrial cancer median survival 8.3 months cervical cancer median survival 8 months leiomyosarcoma weeks secondary to disseminated extracranial primary disease
Uccella S. et al., 2016 [74]	1/18 endometrial cancer	SRS	survival 6 months
Shin H.K. et al., 2016 [75]	14/ovarian cancer 6/endometrial cancer 4/cervical cancer	24 SRS 6 WBRT 3 S 7 Ommaya reservoir insertion	CR in 39 lesions (66.1%) PR in 11 lesions (18.6%) SD in 3 lesions (5.1%) median OS 9.5 months after SRS (range 1–102 months), 6 alive
Matsunaga S. et al, 2016 [76]	33 /ovarian cancer with 147 tumors 37/endometrial cancer with 159 tumors	SRS	local tumor control rates 96.4% at 6 months and 89.9% at 1 year there was no statistically significant difference between ovarian and uterine cancers
Keller A. et al., 2016 [79]	17/ovarian cancer 10/ endometrial cancer 6/cervical cancer	SRS ± WBRT	cervical cancer median survival 17 months endometrial cancer median survival 6 months ovarian cancer median survival 16 months
Kasper E. et al., 2017 [85]	A total of 20 lesions in 8 patients: 1/endometrial cancer 7/ovarian cancer	SRS 3 patients surgical resection 1 patient re-irradiation	the actuarial 1-, 2- and 3-year local control rates were 91, 91 and 76%, respectively the median overall survival time was 29 months
Johnston H. et al., 2017 [87]	2/cervical cancer 6/endometrial cancer 25/ovarian cancer	SRS ± WBRT	local failure at 1 and 2 years for the entire population was 10.4% and 14.3% median overall survival for all patients was 12 months (range 1–77 months)
Bhambhvani H.P. et al., 2021 [100]	30/endometrial cancer	SRS alone 93% SRS + S 36.7%	median survival 5.6 months (range, 1–50.4) vs 15.7 months (range 2.8–58.2 months *p* = 0.17
Beucler N. et al., 2021 [101]	1/endometrial cancer	S + SRS + WBRT	alive 14
Wei Z. et al., 2023 [108]	4/cervical cancer 25/endometrial cancer 21/ovarian cancer	SRS	OS at 6 and 12 months after SRS was 48%, and 44%,
Matsunaga S. et al., 2023 [109]	61 patients with 260 tumors of CC 73 patients with 302 tumors of EC	SRS	local tumor control rates at 6, 12, and 24 months after GKRS were 90.0%, 86.6%, and 78.0% for CC and 92.2%, 87.9%, and 86.4% for EC

CC: cervical carcinoma; CR: complete response; EC: endometrial carcinoma; GKRS: gamma-knife radiosurgery; OS: overall survival; PR: partial response; S: surgery; SD: stable disease; SRS: stereotactic radiotherapy; ST: systemic therapy; WBRT: whole-brain radiotherapy. Systemic treatment for recurrent or metastatic endometrial carcinoma primarily relies on platinum-based combination regimens. The GOG 177 study established combination chemotherapy as the standard of care for advanced or recurrent EC, demonstrating that the TAP regimen—comprising paclitaxel, doxorubicin, and cisplatin—offers significant advantages over doxorubicin and cisplatin alone. These benefits include improved objective response rates (57% vs. 34%; *p* < 0.01), progression-free survival (PFS) (median, 8.3 vs. 5.3 months; *p* < 0.01), and overall survival (median, 15.3 vs. 12.3 months; *p* = 0.037) [147].

## Data Availability

Dataset available on request from the authors.

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
