# Peer review of "Brain Metastasis in Endometrial Cancer: A Systematic Review"

_cancers, 2025, doi:10.3390/cancers17030402_

Round 1

Reviewer 1 Report

Comments and Suggestions for Authors

Brain metastasis from endometrial cancer is an uncommon but serious clinical event, occurring in a small percentage of patients with endometrial cancer.  A review of the literature confirms the rarity of these cases.  The prognosis of patients with these tumors remains poor, with a median survival of 4 months following whole brain radiotherapy and a 12% one-year survival rate.  Surgery alone appears to have limited efficacy, even in patients with single metastases and no extracranial disease.  The integration of immune checkpoint inhibitors, targeted therapies, and stereotactic radiotherapy may offer some hope in managing these cases.  It is unclear why surgery does not provide more benefit particularly in patients with no extracranial disease.  The question remains as to why the survival is so short in these patients; is it due to the intracranial component or due to the natural history of patients with endometrial cancer.  This is an interesting report about a ‘zebra’, but more work needs to be done to understand the issues in these patients.

Author Response

I updated all the tables as requested in editable format and, regarding the possible copyright issues, I created all the figures and tables so there are no problems regarding that. I also updated the references with all the authors' names. I also improved the graphic of all the figures to make them more easily understandable.

Reviewer 2 Report

Comments and Suggestions for Authors

By reviewing a large amount of literature, the authors mainly discussed the impact of different therapeutic strategies on survival of endometrial cancer patient with brain metastases. It is helpful to guide future research and inform clinical decision for the rare condition. In general, the work of this paper is clear and logical, and I suggest that this paper be accepted with major modification.

1A clear description of the quality assessment tool and criteria applied in this study is lacking. The authors need to clarify whether they used a validated scale like the Newcastle-Ottawa Scale or Jadad scale, showing any potential biases that might affect the overall conclusions of the review.

2By searching for literatures about the topic, it has been found that there were already relevant literature reviews on brain metastasis in endometrial cancer. Therefore, the topic of this article is not novel enough. It is recommended that the authors highlight the clinical value of this article. For example, how do the findings of this systematic review translate into actual patient management and what are the potential therapeutic targets or preventive strategies for patients managing?

3The format of this manuscript is chaotic. For example, the layout of some page  needs to be further modified and unified.

Author Response

(The authors gave the same response as above.)

Reviewer 3 Report

Comments and Suggestions for Authors

This systematic review provides a comprehensive and clinically valuable analysis of brain metastases in endometrial cancer (EC), a rare and understudied complication.

-The use of PRISMA guidelines ensures a systematic and transparent methodology, enhancing the study's credibility

suggestions:

1) Consider briefly mentioning how advancements in imaging and systemic therapies may improve early detection and management.

2) Ensure all abbreviations are defined upon first use in tables and figures (e.g., WBRT, SRS).

3) In discussion section,  Expand on the potential role of emerging therapies, such as immunotherapy combined with radiotherapy, and their mechanisms of action in this context.

4) The manuscript is generally well-written, but minor grammatical corrections are needed. For example:

  • “Brain metastases carries a poor prognosis” → “Brain metastases carry a poor prognosis.”

Author Response

(The authors gave the same response as above.)

Round 2

Reviewer 1 Report

Comments and Suggestions for Authors

Brain metastasis from endometrial cancer is an uncommon but serious clinical event, occurring in a small percentage of patients with endometrial cancer.  A review of the literature confirms the rarity of these cases.  The prognosis of patients with these tumors remains poor, with a median survival of 4 months following whole brain radiotherapy and a 12% one-year survival rate.  Surgery alone appears to have limited efficacy, even in patients with single metastases and no extracranial disease.  The integration of immune checkpoint inhibitors, targeted therapies, and stereotactic radiotherapy may offer some hope in managing these cases.  It is unclear why surgery does not provide more benefit particularly in patients with no extracranial disease.  The question remains as to why the survival is so short in these patients; is it due to the intracranial component or due to the natural history of patients with endometrial cancer.  This is an interesting report about a rare event.